# Hemodynamic Adaptations Induced by Short-Term Run Interval Training in College Students

**DOI:** 10.3390/ijerph17134636

**Published:** 2020-06-27

**Authors:** Patricia C. García-Suárez, Iván Rentería, Priscilla García Wong-Avilés, Fernanda Franco-Redona, Luis M. Gómez-Miranda, Jorge A. Aburto-Corona, Eric P. Plaisance, José Moncada-Jiménez, Alberto Jiménez-Maldonado

**Affiliations:** 1Facultad de Deportes, Universidad Autónoma de Baja California-Ensenada, Boulevard Zertuche s/n. Fraccionamiento Valle Dorado, Ensenada 22890, Baja California, Mexico; patricia.garcia@uabc.edu.mx (P.C.G.-S.); irenteria@uabc.edu.mx (I.R.); garcia.priscilla@uabc.edu.mx (P.G.W.-A.); francofer022@gmail.com (F.F.-R.); 2Facultad de Deportes Tijuana, Universidad Autónoma de Baja California, Avenida Maclovio Herrera #4080, Colonia Francisco Villa, Tijuana 22615, Baja California, Mexico; luismariouabc@gmail.com (L.M.G.-M.); jorge.aburto@uabc.edu.mx (J.A.A.-C.); 3Department of Human Studies, University of Alabama at Birmingham, Education Building 901, 13th Street South, Birmingham, AL 35294, USA; plaisep@uab.edu; 4Human Movement Sciences Research Center (CIMOHU), University of Costa Rica, Ave. 31 Pavas, San José 1200, Costa Rica; jose.moncada@ucr.ac.cr

**Keywords:** sprint interval training, VO_2max_, body composition, running, double product, heart rate

## Abstract

Perceived lack of time is one of the most often cited barriers to exercise participation. High intensity interval training has become a popular training modality that incorporates intervals of maximal and low-intensity exercise with a time commitment usually shorter than 30 min. The purpose of this study was to examine the effects of short-term run interval training (RIT) on body composition (BC) and cardiorespiratory responses in undergraduate college students. Nineteen males (21.5 ± 1.6 years) were randomly assigned to a non-exercise control (CON, *n* = 10) or RIT (*n* = 9). Baseline measurements of systolic and diastolic blood pressure, resting heart rate (HRrest), double product (DP) and BC were obtained from both groups. VO_2max_ and running speed associated with VO_2peak_ (sVO_2peak_) were then measured. RIT consisted of three running treadmill sessions per week over 4 weeks (intervals at 100% sVO_2peak_, recovery periods at 40% sVO_2peak_). There were no differences in post-training BC or VO_2_max between groups (*p* > 0.05). HRrest (*p* = 0.006) and DP (*p* ≤ 0.001) were lower in the RIT group compared to CON at completion of the study. RIT lowered HRrest and DP in the absence of appreciable BC and VO_2max_ changes. Thereby, RIT could be an alternative model of training to diminish health-related risk factors in undergraduate college students.

## 1. Introduction

A robust negative relationship exists between weekly amounts of physical activity and cardiometabolic disease morbidity and mortality [1]. The American College of Sports Medicine (ACSM) recommends performing 150 min/week of moderate-intensity or ≥75 min/week of vigorous-intensity physical activity. Although achieving physical activity recommendations is a proven strategy to improve health, undergraduate students, like much of the adult population in developed countries do not meet these recommendations [2,3,4], and report lack of time as the main barrier to achieving these recommendations [5]. In addition, undergraduate students have unhealthy eating habits [6,7] and high levels of psychological stress (e.g., school burnout, anxiety, depression) [8,9,10]. All of these conditions are considered powerful risk markers for cardiovascular disease (CVD) [11,12].

Emerging evidence indicates that high intensity interval exercise training (HIIT), may promote similar benefits as longer duration continuous moderate-intensity exercise which may have important implications for individuals with limited time for exercise [13,14]. For instance, HIIT is characterized by performing relatively short bursts of vigorous activity (e.g., 80% to 100% of the maximal heart rate [HRmax] or 80% to 95% peak oxygen consumption [VO_2peak_]). Intense exercise is interspersed by passive rest or active low-intensity exercise recovery periods (e.g., 40% HRmax) [15,16].

In fact, HIIT is classified based on interval duration, work intensity, session volume and protocol duration [17]. HIIT is exercise modality usually performed in cycle ergometer, motorized treadmill, or running track [13,18,19]. Several reports indicate that HIIT sessions usually consist of 4–8 bouts of high intensity and recovery sessions typically lasting <30 min [20], highlighting the utility of this form of training as it relates to time barriers [21]. Evidence suggests that short-term cycle ergometer interval training interventions (≤4 weeks) improve physiological markers associated with health (e.g., skeletal muscle oxidative capacity and VO_2max_) in male undergraduate students [13,22,23]. The effects of cycle ergometer interval training on body composition have also been reported [24], with a report of lower body fat percentage following 12 cycle-ergometer interval training sessions [24]. Nevertheless, the metabolic demands of the exercise modalities are different in cycling compared to running [25]; thus, previous findings cannot be generalized to run interval training (RIT). In fact, there are only a few studies reporting the effects of short-term high intensity RIT on cardiorespiratory fitness and body composition in undergraduate students [18,26,27]. Moreover, we are unaware of any studies which have examined the impact of short-term RIT on hemodynamic responses in undergraduate students. Therefore, the aim of the present study was to determine the effects of short-term RIT performed on a motorized treadmill on hemodynamic variables, cardiorespiratory fitness, and body composition in young adult undergraduate students. We hypothesized that 12 sessions of RIT would improve cardiorespiratory fitness, hemodynamic, and body composition variables in exercise participants compared to a non-exercise control condition.

## 2. Materials and Methods

### 2.1. Participants

Nineteen physically active (self-reported International Physical Activity Questionnaire [IPAQ] short form: 6819.3 ± 4505.2 METs; minimum value: 1626, maximum value: 15,813) undergraduate male students (age = 21.5 ± 1.6 yr.) from the Sports Faculty of the Universidad Autónoma de Baja California (UABC) participated in the study. Inclusion criteria were: (a) non-smoking, (b) free from any known metabolic or cardiovascular disease, and (c) not taking any medications known to affect hemodynamic or other physiological variables. The procedures were explained to each participant, who agreed to complete the study after reading and signing an institutionally approved informed consent in accordance with the Declaration of Helsinki and previously approved by the Research Ethics Committee of the Escuela de Ciencias de la Salud of UABC (register number 002-2019).

### 2.2. Experimental Design

Participants were given three appointments to the laboratory separated by 24 h. Each participant was instructed to refrain from performing strenuous exercise the day before arrival to the laboratory. The first visit included a complete demonstration of the treadmill (ProForm, Logan, UT, USA), the bioelectrical impedance (BIA) analyzer (Inbody 770, Biospace Corporation, Seoul, South Korea), and the metabolic cart (COSMED Quark CPET, Rome, Italy). Body composition measures were recorded during the second visit, and the graded exercise test (GXT) during the third visit. All laboratory tests were performed between 0900 and 1200 h. Following baseline measurements, participants were assigned by simple randomization to a non-exercise control (CON, *n* = 10) or an exercise (RIT, *n* = 9) group. The participants started the 12 session RIT program 24 h after the GXT. During the training program, participants in both groups were allowed to engage in normal leisure and school physical activities. The CON group only visited the laboratory during the baseline measurements and at the end of the program. Both groups were assessed in the same order as described above 24 h after the last exercise session (Figure 1).

### 2.3. Body Composition Analysis

Height was measured using a stadiometer (Biospace Corporation, Seoul, South Korea) to the nearest ±1 mm. Body composition measures (i.e., body weight [kg], body fat mass [%], muscle mass [kg], leg lean mass [kg], and body mass index [BMI = kg/m^2^]) were recorded by BIA [28]. For quality control, participants were instructed to refrain from eating and drinking for at least 2 h and to void their bladders 60 min prior to performing BIA.

### 2.4. Graded Exercise Test (GXT)

Participants performed a GXT until volitional fatigue to determine fitness level (VO_2max_) [29]. The GXT followed a standard protocol reported before [30,31] with minor modifications. Briefly, the test started with a warm-up run at 5.0 km/h with 1% incline. Then, the treadmill speed was increased by 1 km/h every 2 min until participant’s exhaustion. During the GXT, members of the research team encouraged participants to give their maximal effort. Breath-by-breath samples of expired CO_2_ were collected during the test. The VO_2_ value recorded at the last stage of the GXT was considered the VO_2peak_ (Appendix A). The running speed associated with the VO_2peak_ (sVO_2peak_) was used to design the RIT program. Immediately following the last stage of the GXT, a 4 min cool-down run was performed at 5.0 km/h with 1% incline. The GXT was considered maximal if the participants reached three of the following criteria: (a) respiratory exchange ratio (RER) > 1.10, (b) HRmax within 10 beats of the age-predicted HRmax (220-age), and (c) a VO_2_ plateau despite an increase in workload or running speed [32,33], and/or (d) when the participant requested to stop the test because of volitional exhaustion [34]. The exhaustion time, HRmax and VO_2max_ were recorded at the end of the GXT.

### 2.5. Hemodynamic Responses

Participants rested while sitting comfortably in a chair upon arrival to the laboratory. After 5-min, systolic blood pressure (SBP) and diastolic blood pressure (DBP) were recorded by a digital blood pressure monitor (Omron, Omron Healthcare, Inc., Bannockburn, IL, USA). Resting heart rate (HRrest) was measured by telemetry (Polar FT1, Kempele, Finland). HR recordings during the exercise sessions were registered at the end of each high-intensity bout. Baseline double product (DP [mmHg • bpm/100] = SBP × HRrest) was calculated as an index of myocardial oxygen consumption (MVO_2_) [35,36,37].

### 2.6. Run Interval Training

The RIT program consisted of 12 exercise sessions, with a progressively increasing training volume. Weekly training sessions were performed on Monday, Wednesday, and Friday for four weeks. All training sessions were performed from 0900 to 1300 h in the student’s personal leisure time between classes. The initial three sessions started with a 2 min run warm-up at 40% sVO_2peak_. Then, a high-intensity interval was performed for 2 min at 100% sVO_2peak_, for a total of three high-intensity and low-intensity bouts. The mean intensity was calculated using the equation reported by Billat et al., (2001): (100 + 40)/2 = 70% sVO_2peak_. The high:low interval ratio for these sessions was 2:2 = 1 [38], and the total training time was 12 min. The next four sessions consisted of four cycles of RIT. The high:low interval ratio was 2:1 = 2, for a total duration of 12 min. Finally, for the last five sessions, the number of cycles increased to five (Appendix A), the high:low interval ratio was 2:1 = 2, for a total duration of 15 min. This protocol can be considered as moderate-volume (MV-HIIT), and short duration intervention (ST-HIIT) [17]. To confirm the effect of the workload changes during the intervention, the total distance (km) covered in each exercise session was recorded for volume and intensity. Finally, the HR was monitored continuously throughout all exercise sessions.

### 2.7. Statistical Analysis

Statistical analysis was performed using IBM SPSS version 20.0 (IBM SPSS-Statistics, Armonk, NY, USA). Data are reported as means ± standard deviation (SD). Distribution of the data was assessed with the Shapiro-Wilk’s test. Independent samples *t*-test compared mean baseline anthropometry and body composition between CON and RIT groups. One-way ANOVA was computed to examine mean differences in distance run and HRmax responses in the training sessions S1 (start of program), S6 (middle of the program), and S12 (end of program). Two-way ANOVA with Tukey post-hoc testing was computed to evaluate VO_2max_, hemodynamic, and body composition variables over time (pre vs. post) and groups (CON vs. RIT). Pearson correlation was used to analyze the association among the change (Δ%) in the HRrest and SBP. Effect sizes were computed as Cohen’s *d*, and were interpreted as small (0.2–0.5), moderate (0.5–0.8) and large (>0.8). The 95% confidence intervals (95% CI) around the point estimates are reported. Statistical significance was set a priori at *p* ≤ 0.05.

## 3. Results

Descriptive and inferential statistics of the participants in the CON and RIT groups are presented in Table 1. Following randomization, baseline age, height, weight, BMI, fat mass (%), muscle mass (kg) and lean leg mass (kg) were similar between CON and RIT groups (*p* > 0.05 for all). Distance run during RIT was different between the program stages (*p* = 0.002, *d* = 1.12, 95%CI = 1.11, 1.12). Post-hoc analysis showed that the distance run during S1 (2.8 ± 0.4 km) and S6 (3.0 ± 0.2 km) was similar (*p* = 0.091, 95%CI = 0.0, 0.4). The distance run during S1 was shorter than S12 (3.1 ± 0.2 km, *p* = 0.005, 95%CI = 0.2, 0.6), and the distance run during S12 was longer than S6 (*p* = 0.004, 95%CI = 0.1, 0.2, Figure 2A). These data indicate the positive effects of the workload changes applied during the treatment. Mean HR response to high-intensity bouts was similar between S1, S6 and S12 (*p* = 0.543) (Figure 2B). The mean HR recorded during S1, S6 and S12 were 90% of the HRmax reached during the GXT.

There was no interaction between CON and RIT and measurement times for VO_2max_, exhaustion time, DBP, body weight, BMI, body fat %, muscle mass, and lean leg mass. Although not statistically significant, the RIT intervention showed a strong trend to reduce SBP compared to the post-test of the CON group (*p* = 0.07, *d* = −1.98, 95%CI = −2.11, −1.85) (Table 1). There was an interaction between CON and RIT and measurement times on HRrest (*p* ≤ 0.001, *d* = 2.21, 95%CI = 2.07, 2.35). Follow-up analysis showed that HRrest increased in the CON group (*p* = 0.031, 95%CI = 0.7, 13.1) and decreased in the RIT group (*p* = 0.001, 95%CI = 5.6, 17.4) after the program. Mean HRrest following the program was smaller in the RIT group compared to CON (*p* = 0.006, 95%CI = 4.3, 21.2) (Table 1). A correlation was found between ΔHRrest and ΔSBP (*r* = 0.52, *p* = 0.02; *d* = 0.27, 95%CI = 0.09, 0.79) (Figure 3). There was an interaction between CON and RIT and measurement times on DP (*p* ≤ 0.001, *d* = 2.06, 95%CI = −19.90, 24.02). Follow-up analysis showed that DP increased in the CON group (*p* = 0.012, 95%CI = 3.2, 22.7) and decreased in the RIT group (*p* = 0.005, 95%CI = 4.8, 23.4) after the program. Mean DP was lower in the RIT group compared to CON following the program (*p* ≤ 0.001, 95%CI = 12.1, 34.3) (Table 1).

## 4. Discussion

The aim of the current study was to determine the effects of short-term RIT on cardiorespiratory fitness, hemodynamic responses, and body composition in healthy undergraduate students. The main finding of the study was a significant reduction in DP and HRrest following 4 weeks of RIT. However, the 12 sessions of RIT did not modify fitness levels or body composition.

The total training volume throughout all sessions averaged ≤15 min, which agrees with the training duration suggested for HIIT or SIT [39]. The HR reached during each high-intensity bout was approximately 90% of HRmax. Since the high-intensity bouts were performed at 100% of sVO_2peak_, our results show that monitoring HR is not a good indicator for controlling the intensity of in interval training, this phenomena was also previously reported in healthy young adults [40,41,42]. In contrast to previous reports, VO_2max_ was not increased following short-term training compared to CON [18,27]. Participants in the current study showed higher baseline VO_2max_ values than those previously reported [18,27], which could partially explain the lack of significant changes in aerobic power. Others have suggested a similar hypothesis [41]. Moreover, although the studies that reported an improvement in VO_2max_ directly had comparable intervention durations (12 sessions over 4 weeks) compared with the current study, the sessions’ design were completely different (short-interval vs. moderate interval in the current study) [17,18,27]. These data suggest that interval duration is relevant to increase cardiorespiratory fitness in young adults [24,43]. Finally, although we did not examine peripheral adaptations, other reports show that 18 sessions of interval training improved VO_2max_ in undergraduate students by inducing skeletal muscle adaptations [26,44].

In this study, we report that a short-term RIT protocol reduced HRrest; these data are in agreement with other authors that used short-term interval training in cycling exercises [45]. In the current study, heart rate variability was not measured; thus, we cannot determine whether the lower HRrest resulted from an elevated vagal activity following training. Others have previously demonstrated that the physiological mechanism induced by short-term interval training responsible for reducing HRrest is an intrinsic adaptation of the sinoatrial node rather autonomic activity changes [45]. Therefore, it is possible that similar physiological adaptations might have occurred in our participants to reduce HRrest. Contrary, the CON group increased HRrest, despite this outcome, their values match with previous HRrest data reported in healthy young adults [46,47,48].

The RIT program designed for the present study produced a decrease in SBP compared to CON, yet, this finding did not reach statistical significance. The trend observed agrees with a recent report that showed the efficacy of short-term interval training to reduce blood pressure [45]; additionally, a recent meta-analysis identified a significant benefit of interval training to improve daytime resting blood pressure [12]. It has been reported that peripheral vascular adaptations are the main interval training-related mechanisms responsible for reducing resting blood pressure [49,50]. Cross-sectional studies have reported that abnormal blood pressure values (e.g., pre-hypertension, hypertension) are common in young adult populations [51,52]. In undergraduate students, high blood pressure is a consequence of higher sympathetic tone [53], and our data are in agreement with this physiological mechanism, where a lower HRrest was associated with lower SBP (Figure 3). Therefore, RIT could be an effective strategy to regulate resting blood pressure in a predisposed population of students characterized by having high-levels of sedentary time, lack of time for physical activity/exercise, and high levels of school burnout [53,54]. In addition, resting DP changed differentially in the RIT and CON groups. DB was lower at the end of the study in the RIT and higher in the CON (within-group analysis), and DP was lower in the RIT compared to the CON following intervention (between-group analysis). These data are in concordance with a lower HRrest [55] (Table 1). The DP is a non-invasive method to estimate MVO_2_ [56]. Others have reported lower resting DP after long-term resistance training in hypertensive women [57]; however, the effects of short-term interval training on resting and exercise DP have not been reported. The usefulness of DP at rest has been previously suggested in the context of MVO_2_ [58,59] in that a higher DP at rest was more strongly associated with cardiovascular disease (CVD) mortality, and non-CVD mortality than other cardiovascular biomarkers (e.g., SBP, DBP, HRrest) [53]. A low MVO_2_ is an indicator of improvement in left ventricular relaxation, and changes in myocardial substrate metabolism [60,61]. Although we did not evaluate MVO_2_ directly, our data suggest that 12 RIT sessions may have induced metabolic changes in the heart that lead to lower estimates of cardiac muscle VO_2_.

In the present study, body composition did not change with 12 sessions of RIT. Our results are in agreement with others who employed a similar short-term running interval training protocols [18]. In contrast, studies using longer interval training programs (≥6 weeks) have reported positive changes in body composition [19,26,62]. These findings suggest that the length of the training program is a relevant variable to induce positive changes in body composition. It is worth noting that in the current study, the calorie intake and composition were not controlled, which may have dampened the training effects on body composition [63,64].

A potential limitation of the current study was that we only included males. Additional short- and long-term studies examining both men and women in parallel cohorts will be important in future studies to further examine hemodynamic and cardiorespiratory responses to exercise. In addition, participants in the current study had high baseline fitness levels which along with the short duration of the study could have limited the magnitude of improvement in both fitness and body composition. In addition, we did not monitor changes in physical activity levels throughout the study; however, participants were instructed to maintain unchanged their physical activity levels throughout the duration of the study. Perhaps accelerometry might be used in future studies to accurately confirm habitual physical activity levels. Finally, this study did not determine ventilatory thresholds (VT), a variable used to assess functional capacity in individuals [65]. Scientific evidence indicates significant changes in VT after long and mid-term interval training [27,66,67]; therefore, we do not discard the possibility that RIT could modify VT in the participants.

## 5. Conclusions

Twelve RIT sessions performed over 4 weeks, and executed during short breaks of daily academic activities significantly decreased HRrest and an estimate of MVO_2_ in male undergraduate students with high cardiorespiratory fitness levels. In contrast, the exercise intervention did not modify VO_2max_ and body composition. Based on previous studies using longer exercise duration, it seems plausible that a threshold training length is needed to observe positive changes in cardiorespiratory fitness and body composition. However, reduced heart rate and DP indicate that metabolic changes occur rapidly even in the absence of reductions in body weight or adiposity (at least in males). Further studies are needed to examine the time course of effects of RIT on cardiorespiratory fitness and body composition in male and female undergraduate students.

## Figures and Tables

**Figure 1 ijerph-17-04636-f001:**
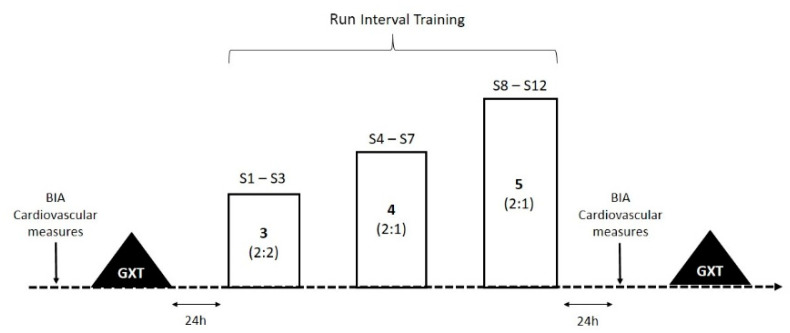
Flow chart of study procedures. Training sessions are depicted as open bars and the numbers inside bars indicate the interval ratio (high–low intensity) and bold numbers represent the 2 min “all out” sprints performed during each session. Cardiovascular measures, bioelectrical impendence (BIA) and graded exercise tests (GXT) were performed 24 h before and after the RIT protocol.

**Figure 2 ijerph-17-04636-f002:**
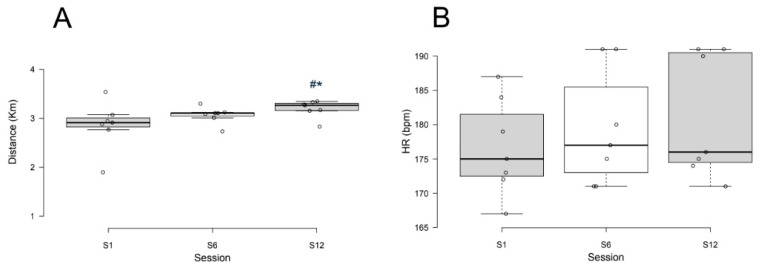
Distance run during the first session of training (S1), at the middle of the protocol (S6), and during the last session of RIT (S12) (**A**). Heart rate recorded during the RIT sessions (**B**). * *p* = 0.005 S1 vs. S12; ^#^
*p* = 0.004 S6 vs. S12.

**Figure 3 ijerph-17-04636-f003:**
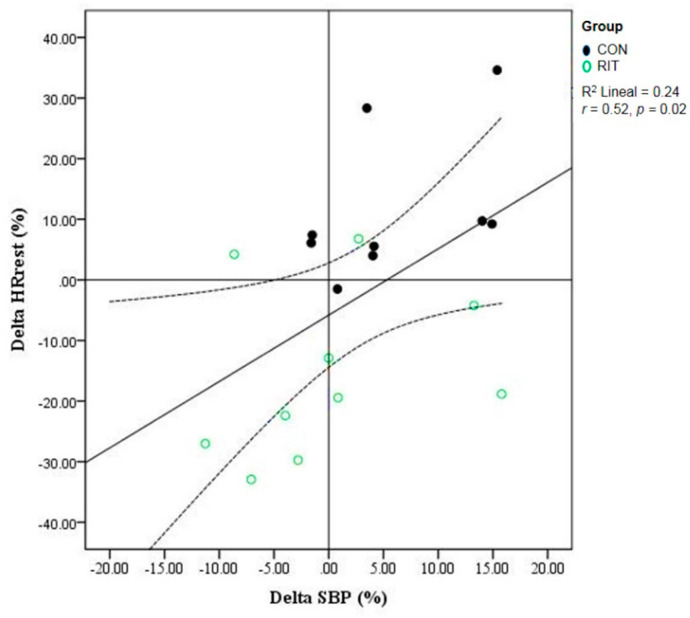
Scatter plot for the correlation analysis between ΔHRrest and ΔSBP. Δ = ([Post-intervention–Pre-intervention values]/Pre-intervention values]) × 100. Dashed lines are the 95% CI around the regression line. *r* = 0.52, *p* = 0.02.

**Table 1 ijerph-17-04636-t001:** Inferential statistics by experimental group and measurement time.

Variable	CON (*n* = 10)		RIT (*n* = 9)	
	Pre	Post	Pre	Post
Weight (kg)	65.7 ± 5.9	65.9 ± 5.9	68.4 ± 7.0	68.0 ± 7.3
BMI (kg·m^2^)	22.4 ± 2.1	22.5 ± 2.1	23.2 ± 2.0	23.1 ± 2.1
Body fat (%)	14.9 ± 3.2	14.9 ± 3.9	14.5 ± 4.0	15.1 ± 4.0
Muscle mass (kg)	31.6 ± 2.8	31.8 ± 3.1	33.3 ± 4.0	32.8 ± 4.0
Lean leg mass (kg)	17.5 ± 2.1	17.5 ± 2.1	16.6 ± 3.6	17.4 ± 2.7
Exhaustion time (min)	19.9 ±2.4	22.4 ± 4.1	17.7 ± 3.8	19.6 ± 2.4
HR_rest_ (bpm)	64.4 ± 10.4	71.3 ± 10.0 *	70.1 ± 6.5	58.6 ± 7.4 ^#§^
SBP (mmHg)	120.2 ± 7.8	127.0 ± 5.1	115.5 ± 15.2	114.3 ± 7.5
DBP (mmHg)	70.7 ± 2.8	74.6 ± 10.4	66.4 ± 11.8	64.1 ± 7.5
DP (mmHg·bpm/100)	77.4 ± 13.4	90.4 ± 11.5 ^¥^	81.3 ± 15.3	67.2 ± 11.4 ^¶¤^
VO_2max_ (mL·kg^−1^·min^−1^)	53.7 ± 9.9	51.4 ± 6.4	51.3 ± 3.8	52.3 ± 5.2

CON = control group; RIT = run interval training group; BMI = body mass index; HR_rest_ = resting heart rate; SBP = systolic blood pressure; DBP = diastolic blood pressure; DP = double product; VO_2max_ = maximal oxygen uptake. * *p* = 0.031 indicates significant differences between post vs. pre in the CON group; ^#^
*p* = 0.001 indicates significant differences between post vs. pre in the RIT group; ^§^
*p* = 0.006 indicates significant differences between CON vs. RIT in the post-test measurement; ^¥^
*p* = 0.012 indicates significant differences between post vs. pre in the CON group; ^¶^
*p* = 0.005 indicates significant differences between post vs. pre in the RIT group; ^¤^
*p* ≤ 0.001 indicates significant differences between CON vs. RIT groups in the post-test measurement.

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
