# Peer review of "Hemodynamic Adaptations Induced by Short-Term Run Interval Training in College Students"

_ijerph, 2020, doi:10.3390/ijerph17134636_

Round 1
Reviewer 1 Report
Overall I think this is a good, interesting, and useful study. However, there are a few adjustments that will make this manuscript communicate the findings more effectively.
A big-picture question I have is, should it be called SIT?
In lines 49-55 you define HIIT and SIT, and this protocol fits more into HIIT. As you state at 135: ‘This protocol can be considered as moderate-volume (MV-HIIT), and short duration intervention (ST-HIIT)” You also cite several studies (14, 37) that define SIT as 30-s bouts of supra-maximal exercise. Why are you calling it SIT then? This is misleading to the reader and should be changed.
The referencing format needs to be corrected throughout. For example, [2]–[4] should be [2-4] and [11], [12] should be [11, 12].
25 - Please include that the controls were not exercising i.e. “… were randomly assigned to non-exercise control”
29/30 - This sentence might read better if it said: “There were no differences in pre or post- training BC or VO2mx between groups (p>0.05).”
46 - This should be a new paragraph starting with “Emerging…”, which can then be connected with the paragraph at line 49.
62 - Remove the word “completely”
73 - Participants were from both Ensenada and Tijuana campuses. Was testing done in the same lab? If so, this is probably not necessary to state which campuses they were from.
82 - Were there any instructions to the control group regarding what type of exercise or activity they could do?
116 - Was it a single BP reading or multiple/repeated readings?
https://www.ncbi.nlm.nih.gov/pmc/articles/PMC4170363/
Results - Can you also include the 95% CI for the effect sizes?
I’m confused about the distance run as shown in Fig 1. If running speed associated with the VO2peak (sVO2peak) was used to design the SIT program, each interval was pre-set according to their baseline test, correct? And then you increased the volume each week. Of course there will be an increase in distance covered? Not sure why that is needed to be shown in a figure? Perhaps I am missing something. Maybe it would be better to also report avg (+/- SD) running speed for each interval since it wouldn’t have changed through the study? This could be instead of, or in addition to Fig. 1A?
Although there were no changes in vo2max, I think it would be valuable to also report the pre vs. post ventilatory thresholds. Did you calculate them?
167 - Can you check this calculation —> “The intervention showed a strong trend to decrease SBP (p = 0.07, d = 1.98) (Table 1).”
That seems like a very high effect size, especially with a non-significant p-value. I calculated it and got a lower number.
195 - Please add “following 4 wk of SIT" - “The main finding of the study was a significant reduction in DP and HRrest following 4 wk of SIT.
196 - “However, the twelve sessions of run SIT did not modify fitness levels or body composition.” Perhaps ventilatory threshold was affected?
201 - it is unclear why you are referencing [39-41] here?
202 - “Participants in the current study showed higher baseline VO2max values than those reported [16], [25], which could explain the lack of significant changes in aerobic power.”
This doesn’t sound as plausible as differences in training protocol. What about other studies that show improvements in similar or fitter subjects?
https://www.ncbi.nlm.nih.gov/pubmed/27580145/
https://www.tandfonline.com/doi/full/10.1080/17461391.2017.1383515
207 - “These data suggest that interval duration is relevant to increase cardiorespiratory fitness in young adults.” Agreed!!
391 - weird typo on the word “ In fl uence”
Author Response
May 15th, 2020
Prof. Dr. Paul B. Tchounwou
Editors-in-Chief
International Journal of Environmental Research and Public Health
Subject: Revision of manuscript ID ijerph-799869
We would like to thank to the reviewers for their comments. Please find below a point-by-point response to each comment raised by the reviewers of our manuscript. We discussed their comments and agree with most of the observations. We made the necessary revisions and included new literature to support our ideas. We strongly believe that we have improved the quality of the manuscript.
Thank you,
Alberto Jiménez-Maldonado, Ph.D.
Corresponding author
Facultad de Deportes
Universidad Autónoma de Baja California
We show the original comment/question in boldface followed by our response and the proposed changes to the manuscript in blue.
In the original manuscript we show the changes tracks in blue for any addition and red (strikethrough) for any deletion.
REVIEWER 1
- Overall I think this is a good, interesting, and useful study. However, there are a few adjustments that will make this manuscript communicate the findings more effectively.
Answer: We thank the reviewer for this comment and for the time devoted to scrutinize our manuscript and for the valuable comments, suggestions and corrections requested. We have made the suggested adjustments and we believe that the manuscript reads better now.
- A big-picture question I have is, should it be called SIT? In lines 49-55 you define HIIT and SIT, and this protocol fits more into HIIT. As you state at 135: ‘This protocol can be considered as moderate-volume (MV-HIIT), and short duration intervention (ST-HIIT)” You also cite several studies (14, 37) that define SIT as 30-s bouts of supra-maximal exercise. Why are you calling it SIT then? This is misleading to the reader and should be changed.
Answer: We thank the reviewer for this observation.
Changes: We adjust the term SIT for Run interval training (RIT) throughout the manuscript.
- The referencing format needs to be corrected throughout. For example, [2]–[4] should be [2-4] and [11], [12] should be [11, 12].
Answer: We thank the reviewer for this comment.
Changes: The references format were corrected
- 25 - Please include that the controls were not exercising i.e. “… were randomly assigned to non-exercise control”
Answer: We thank the reviewer for this suggestion.
Change: The correction was made (Ln 99 of the revised version)
- 29/30 - This sentence might read better if it said: “There were no differences in pre or post- training BC or VO2mx between groups (p>0.05).”
Answer: We thank the reviewer for this suggestion.
Change: The correction was made (Ln 31 of the revised manuscript version)
- 46 - This should be a new paragraph starting with “Emerging…”, which can then be connected with the paragraph at line 49.
Answer: We thank the reviewer for this suggestion.
Changes: The suggestion was attended (Ln 49-55 of the revised manuscript version)
- 62 - Remove the word “completely”
Answer: We thank the reviewer for this observation
Changes: The correction was made (Ln 68 of the revised manuscript version)
- 73 - Participants were from both Ensenada and Tijuana campuses. Was testing done in the same lab? If so, this is probably not necessary to state which campuses they were from.
Answer: We thank the reviewer for this observation. Yes, the testing was performed in the same lab
Changes: The suggestion has been attended (Ln 81-83 of the revised manuscript version)
- 82 - Were there any instructions to the control group regarding what type of exercise or activity they could do?
Answer: We thank the reviewer for this question. In lines 100-102 of the revised manuscript version), we describe the instructions made to the participants to maintain their leisure activities.
- 116 - Was it a single BP reading or multiple/repeated readings?
https://www.ncbi.nlm.nih.gov/pmc/articles/PMC4170363/
Answer: We thank the reviewer for this observation. The BP measure was performed once in every testing (same time of day) by the same tester.
- Results - Can you also include the 95% CI for the effect sizes?
Answer: We thank the reviewer for this question.
Changes: The information has been included in the results section.
- I’m confused about the distance run as shown in Fig 1. If running speed associated with the VO2peak (sVO2peak) was used to design the SIT program, each interval was pre-set according to their baseline test, correct? And then you increased the volume each week. Of course there will be an increase in distance covered? Not sure why that is needed to be shown in a figure? Perhaps I am missing something. Maybe it would be better to also report avg (+/- SD) running speed for each interval since it wouldn’t have changed through the study? This could be instead of, or in addition to Fig. 1A?
Answer: We thank the reviewer for this question.
Changes: We described further details regarding the report of covered distance in Methods (ln 150-154) and Results (ln 177/178) sections.
- Although there were no changes in vo2max, I think it would be valuable to also report the pre vs. post ventilatory thresholds. Did you calculate them?
Answer: We thank the reviewer for this question. Unfortunately, we updated desktop computers between 2018 and 2020 and the data is missing. We apologize for being unable to reply to the reviewer’s comment appropriately.
Changes: As mentioned above, we included this issue (ventilatory thresholds) in the Discussion section (Ln 282-285 of the revised manuscript version).
- 167 - Can you check this calculation —> “The intervention showed a strong trend to decrease SBP (p = 0.07, d = 1.98) (Table 1).” That seems like a very high effect size, especially with a non-significant p-value. I calculated it and got a lower number.
Answer: We thank the reviewer for this observation.
We used the following formula to compute the between-group Cohen's d: (M2 - M1) ⁄ SDpooled, where SDpooled = √((SD12 + SD22) ⁄ 2). The post-test mean ± SD values were: SIT = 114.3 ± 7.5 and CON = 127.0 ± 5.1. Thus, d = (114.3 - 127) ⁄ 6.41 = -1.98 (95%CI = -2.11, -1.85). The reviewer probably obtained a lower ES by computing the within-group ES as follows:
For SIT, d = (114.3 - 115.5) ⁄ 11.9 = -0.10 (95%CI = -0.35, 0.14).
For CON, d = (127 - 120.2) ⁄ 6.6 = 1.03 (95%CI = 0.90, 1.16)
We acknowledge that we missed the negative sign (-) in our original ES and now it reads correct (-1.98).
- 195 - Please add “following 4 wk of SIT" - “The main finding of the study was a significant reduction in DP and HRrest following 4 wk of SIT.
Answer: We thank the reviewer for this suggestion.
Changes: The correction has been made (Ln 217/218 of the revised manuscript version).
- 196 - “However, the twelve sessions of run SIT did not modify fitness levels or body composition.” Perhaps ventilatory threshold was affected?
Answer: We thank the reviewer for this question. Unfortunately, we updated desktop computers between 2018 and 2020 and the data is missing. We apologize for being unable to reply to the reviewer’s comment appropriately.
Changes: As mentioned above, we included this issue (ventilatory thresholds) in the Discussion section (Ln 282-285 of the revised manuscript version)..
- 201 - it is unclear why you are referencing [39-41] here?
Answer: We thank the reviewer for this question.
Changes: To make sense of the references, we included further information in the lines indicated (Ln 224/225 of the revised manuscript version).
202 - “Participants in the current study showed higher baseline VO2max values than those reported [16], [25], which could explain the lack of significant changes in aerobic power.” This doesn’t sound as plausible as differences in training protocol. What about other studies that show improvements in similar or fitter subjects?
https://www.ncbi.nlm.nih.gov/pubmed/27580145/
https://www.tandfonline.com/doi/full/10.1080/17461391.2017.1383515
Answer: We thank the reviewer for this comment.
Changes: We read the suggested studies and discussed with more detail the lack effect of the interval training on VO2max (Ln 225/235 of the revised manuscript version).
- 207 - “These data suggest that interval duration is relevant to increase cardiorespiratory fitness in young adults.” Agreed!!
Answer: Thank you.
- 391 - weird typo on the word “ In fl uence”
Answer: Thank you for this observation. The spacing was off and now it reads fine.

Reviewer 2 Report
It was with great pleasure that I was invited to complete this review. I thought that the hypothesis was sound, and intriguing. I liked how the Intro was structured, but had issues regarding the presentation of the methods. My main concern is lack of power. The authors provide CIs, which is great, but the low number of subjects makes me nervous about the ability to interpret these data.
Major concerns:
I had issues with the continual change in VO2max versus peak but no explanation how they are simliar or different. The authors set to determine VO2max, but settled for VO2 peak line page 3, 103 then 109. This is challenging as the authors use both peak and max interchangeably. But are they? The authors didn't present an argument which really limits the readability in my opinion.
Minor concerns:
Page 2, line 48. Emerging studies but no references.
Page 2, line 49-52. Paragraph is only 2 sentences, maybe incorporate into above paragraph.
Author Response
May 15th, 2020
Prof. Dr. Paul B. Tchounwou
Editors-in-Chief
International Journal of Environmental Research and Public Health
Subject: Revision of manuscript ID ijerph-799869
We would like to thank to the reviewers for their comments. Please find below a point-by-point response to each comment raised by the reviewers of our manuscript. We discussed their comments and agree with most of the observations. We made the necessary revisions and included new literature to support our ideas. We strongly believe that we have improved the quality of the manuscript.
Thank you,
Alberto Jiménez-Maldonado, Ph.D.
Corresponding author
Facultad de Deportes
Universidad Autónoma de Baja California
We show the original comment/question in boldface followed by our response and the proposed changes to the manuscript in blue.
In the original manuscript we show the changes tracks in blue for any addition and red (strikethrough) for any deletion.
REVIEWER 2
It was with great pleasure that I was invited to complete this review. I thought that the hypothesis was sound, and intriguing. I liked how the Intro was structured, but had issues regarding the presentation of the methods. My main concern is lack of power. The authors provide CIs, which is great, but the low number of subjects makes me nervous about the ability to interpret these data.
Answer: We thank the reviewer for the time devoted to scrutinize our manuscript and for the valuable comments, suggestions and corrections requested.
Major concerns:
- I had issues with the continual change in VO2max versus peak but no explanation how they are similar or different. The authors set to determine VO2max, but settled for VO2 peak line page 3, 103 then 109. This is challenging as the authors use both peak and max interchangeably. But are they? The authors didn't present an argument which really limits the readability in my opinion.
Answer: We thank the reviewer for this comment. In the GXT section, the VO2peak was used to design the interval training protocol. Meanwhile, the VO2max was determined to describe the fitness level of the subjects. Both terms were not used interchangeably as both values were achieved at maximal effort by the participants (New supplementary figure attached).
Minor concerns:
- Page 2, line 48. Emerging studies but no references.
Answer: We thank the reviewer for this observation.
Changes: References were included in this statement (Ln 49/50 of the revised manuscript version).
- Page 2, line 49-52. Paragraph is only 2 sentences, maybe incorporate into above paragraph.
Answer: We thank the reviewer for this comment.
Changes: Based on this and other reviewer’s comment, we have incorporated a previous sentence and made the appropriate adjustments to the text (Ln 49/55 of the revised manuscript version).

Reviewer 3 Report
The authors have measured the hemodynamic response, cardiorespiratory fitness, and body composition in young adult undergraduate students to 12 sessions of SIT. The study indicates SIT lowered resting HR and thus double product, suggesting SIT is beneficial for lowering hemodynamic stress in this population. The study was well designed and written. Below are some comments that the authors need to consider to improve the method description and discussion of their manuscript.
Major comments
- The authors are encouraged to include the supplementary figure in the main article because it is crucial to understand the format of the study.
- How were participants randomised to each group? Exercise capacity?
- If the study is designed to determine the effectiveness of SIT for undergraduate students who do not have time to complete normal training and are under continued stress, surely a quantitative or at least qualitative measure of stress should have been conducted prior to, during and after the study. This would have ensured stress levels did not impact changes in blood pressure.
- Why was there no attempt to complete a crossover design?
- The authors need to provide data about sex differences in the study. Was there a difference between males and females? Now, after getting to the limitations section of the study, do I see that only males were included. This should be at the start of the manuscript.
- Consideration needs to be given to the body composition data of this study given that the BIA can overestimate body fat percentages in individuals with a high lean mass.
- Statistics – the authors have neglected a “small” effect for Cohen’s d. Small should be 0.2-0.5, moderate 0.5-0.8 and large >0.8.
- Were activity levels monitored during periods between sessions to ensure no difference both within each group and between groups occurred?
- I am unsure how there was a “trend” for SIT to decrease SBP. The table indicates 115.5 à3. How is this a trend? The discussion needs to be edited accordingly (line 219).
- The interpretation of DP for the SIT group is warranted given the massive change in HR at rest, but care needs to be taken to assume the same extent of change between groups. This is because if the CON group were allowed to participate in normal activity (they were physically active), then it seems unusual that their resting HR would increase by so much and dramatically change DP.
Minor comments
- Line 74 - Can the authors please detail what “physically active” is.
- This study measured changes in blood pressure following SIT in 21-27 years olds (https://www.nrcresearchpress.com/doi/abs/10.1139/apnm-2012-0136#.XqtCmagzaUk)
- Line 201 – VO2 peak not max, correct?
- Double product needs to be explicitly detailed in the abstract.
Author Response
May 15th, 2020
Prof. Dr. Paul B. Tchounwou
Editors-in-Chief
International Journal of Environmental Research and Public Health
Subject: Revision of manuscript ID ijerph-799869
We would like to thank to the reviewers for their comments. Please find below a point-by-point response to each comment raised by the reviewers of our manuscript. We discussed their comments and agree with most of the observations. We made the necessary revisions and included new literature to support our ideas. We strongly believe that we have improved the quality of the manuscript.
Thank you,
Alberto Jiménez-Maldonado, Ph.D.
Corresponding author
Facultad de Deportes
Universidad Autónoma de Baja California
REVIEWER 3
The authors have measured the hemodynamic response, cardiorespiratory fitness, and body composition in young adult undergraduate students to 12 sessions of SIT. The study indicates SIT lowered resting HR and thus double product, suggesting SIT is beneficial for lowering hemodynamic stress in this population. The study was well designed and written. Below are some comments that the authors need to consider to improve the method description and discussion of their manuscript.
Answer: We thank the reviewer for the time devoted to scrutinize our manuscript and for the valuable comments, suggestions and corrections requested. We have made the suggested adjustments and we believe that the manuscript reads better now.
Major comments
- The authors are encouraged to include the supplementary figure in the main article because it is crucial to understand the format of the study.
Answer: We thank the reviewer for this observation.
Changes: We moved the supplementary figure to the main article as was suggested by the reviewer (Ln 104-109 of the revised manuscript version).
- How were participants randomised to each group? Exercise capacity?
Answer: We thank the reviewer for this comment. The groups were assigned by simple randomization.
Changes: We add this information in the experimental design section (Ln 104-109 of the revised manuscript version) (Ln 99-100 of the revised manuscript version).
- If the study is designed to determine the effectiveness of SIT for undergraduate students who do not have time to complete normal training and are under continued stress, surely a quantitative or at least qualitative measure of stress should have been conducted prior to, during and after the study. This would have ensured stress levels did not impact changes in blood pressure.
Answer: We thank the reviewer for this comment. We agree with the reviewer that the psychological stress plays a role on the blood pressure [1, 2] and VO2max responses. It is mean psychological stress is inversely associated with higher VO2max levels [3]. In our study the participants (both control and experimental) showed similar VO2max levels before and after treatment, suggesting that the psychological stress was not different.
References
- May, R.W.; Sanchez-Gonzalez, M.A.; Fincham, F.D. School burnout: Increased sympathetic vasomotor tone and attenuated ambulatory diurnal blood pressure variability in young adult women. Stress, 2014, 18(1), 11–19. doi: 10.3109/10253890.2014.969703.
- May, R.W.; Seibert, G.S.; Sanchez-Gonzalez, M.A.; Fincham, F.D. School burnout and heart rate variability: risk of cardiovascular disease and hypertension in young adult females. Stress, 2017, 21(3), 211–216. doi: 10.1080/10253890.2018.1433161.
- Ritvanen, T. et al. Effect of aerobic fitness on the physiological stress responses at work. International Journal of Occupational Medicine and Public Health, 2007, 20(1), 1-8. doi: 2478/v10001-007-0005-5.
- Why was there no attempt to complete a crossover design?
Answer: We thank the reviewer for this comment. It is a very good and interesting suggestion that we will consider in future studies. In the current study we worked with pre and post-test with control group design.
- The authors need to provide data about sex differences in the study. Was there a difference between males and females? Now, after getting to the limitations section of the study, do I see that only males were included. This should be at the start of the manuscript.
Answer: We thank the reviewer for your observation.
Changes: In the line 25 of the abstract (of the revised manuscript version) we stated that male participants were included in the study. We have also included the word males in line 75 of the Methods section (Ln 82 of the revised manuscript version)..
- Consideration needs to be given to the body composition data of this study given that the BIA can overestimate body fat percentages in individuals with a high lean mass.
Answer: We thank the reviewer for this comment. Reports indicate that the BIA method is a valid tool for the assessments of total body and segmental body composition contrasted against the gold standard (DXA) [1]. In addition, the repeated measures design and the statistical analysis allowed us to determine not only within- but also between-subjects comparisons. Finally, we would like indicate that authors have demonstrated the risk to overestimate the body fat by BIA is present in obese and overweight population [2, 3]. Opposite, the current work was performed in normal weight population
- H. Y. Ling et al., “Accuracy of direct segmental multi-frequency bioimpedance analysis in the assessment of total body and segmental body composition in middle-aged adult population,” Clin. Nutr., vol. 30, no. 5, pp. 610–615, 2011, doi: 10.1016/j.clnu.2011.04.001.
- Pimentel, G. D., Bernhard, A. B., Frezza, M. R. P., Rinaldi, A. E. M., & Burini, R. C. (2010). Bioelectric impedance overestimates the body fat in overweight and underestimates in Brazilian obese women: a comparation with Segal equation 1. Nutricion hospitalaria.
- Li Y-C, Li C-I, Lin W-Y, Liu C-S, Hsu H-S, et al. (2013) Percentage of Body Fat Assessment Using Bioelectrical Impedance Analysis and Dual-Energy X-ray Absorptiometry in a Weight Loss Program for Obese or Overweight Chinese Adults. PLoS ONE 8(4): e58272. doi:10.1371/journal.pone.0058272
- Statistics – the authors have neglected a “small” effect for Cohen’s d. Small should be 0.2-0.5, moderate 0.5-0.8 and large >0.8.
Answer: We thank the reviewer for this observation.
Changes: We made the requested correction.
- Were activity levels monitored during periods between sessions to ensure no difference both within each group and between groups occurred?
Answer: We appreciate the reviewer for this observation. We did not monitor activity levels during the length of the study. However, the data of VO2max and exhaustion time (table 1) after the interval training indicate that physical activity habits did not differ between and within groups.
- I am unsure how there was a “trend” for SIT to decrease SBP. The table indicates 115.5 à3. How is this a trend? The discussion needs to be edited accordingly (line 219).
Answer: We thank the reviewer for this comment. Within the context of our findings, a p = 0.07 was considered a trend for the reduction in SBP.
Changes: We rephrased lines 186-88 (revised manuscript version) to clarify this issue, which was also mentioned in lines 244-245 of the current version of the manuscript
- The interpretation of DP for the SIT group is warranted given the massive change in HR at rest, but care needs to be taken to assume the same extent of change between groups. This is because if the CON group were allowed to participate in normal activity (they were physically active), then it seems unusual that their resting HR would increase by so much and dramatically change DP.
Answer: We thank the reviewer for this comment. Despite that the CON group increased HRrest, their values matches with previous HRrest data showed in healthy young adults [1, 2, 3].
- Hajsadeghi et al., Effects of energy drinks on blood pressure, heart rate, and electrocardiographic parameters: An experimental study on healthy young adults. 2015.
- Zhang, D., Shen, X., & Qi, X. (2016). Resting heart rate and all-cause and cardiovascular mortality in the general population: a meta-analysis. Cmaj, 188(3), E53-E63.
- Costa, A., Bosone, D., Zoppi, A., D'Angelo, A., Ghiotto, N., Guaschino, E., ... & Fogari, R. (2018). Effect of Diazepam on 24-Hour Blood Pressure and Heart Rate in Healthy Young Volunteers. Pharmacology, 101(1-2), 86-91.
Minor comments
- Line 74 - Can the authors please detail what “physically active” is.
Answer: We thank the reviewer for this comment.
Changes: We included new information to characterize the physically active term (Ln 81-82 of the revised manuscript version).
- This study measured changes in blood pressure following SIT in 21-27 years olds (https://www.nrcresearchpress.com/doi/abs/10.1139/apnm-2012-0136#.XqtCmagzaUk)
Answer: We thank the reviewer for this comment. We would like to indicate that the reference suggested by the reviewer used acute interval exercise. On the contrary, our study determined the impact of a short-term of interval training on hemodynamic variables. Therefore, it is not possible to discuss the main outcomes of the suggested reference in our study.
- Line 201 – VO2 peak not max, correct?
Answer: We thank the reviewer for this observation. The term VO2max is correct
- Double product needs to be explicitly detailed in the abstract.
Answer: We thank the reviewer for this comment. We understand the relevance of the comment; however, we considered mentioning all the hemodynamic and fitness variables to present a broad abstract. In addition, we have a 200-word count limit for the abstract. The last situation is the main issue to describe a detailed information regarding DP in the abstract.

Round 2
Reviewer 3 Report
The authors have responded to my original comments. I have a few follow up comments related to my original one.
Comments
- Line 222 – I guess the authors were lucky in that the randomization method resulted in no differences between groups for baseline characteristics.
- Original comment 4 – the authors have not answered “why” they did not choose this design. It is a more powerful option and not addressed in the response.
- Original comment 8 – Yes, the authors are mostly correct. However, RIT did not change VO2max but it did change other parameters. Therefore, activity outside the scope of the study may have influenced results other than VO2max. The authors need to discuss this as a limitation
- Original comment 10 – the authors have not considered the comment with respect to “this” study. Changes in HRrest massively influenced the outcome. Per my original comment, the authors need to discuss this matter.
- All the changed text does not align with the response to reviewers. For example, comment 9, author’s state that changes in SIT SBP were discussed in 186-188 and 244-245, except there is no line 245.
Author Response
May 19th, 2020
Prof. Dr. Paul B. Tchounwou
Editors-in-Chief
International Journal of Environmental Research and Public Health
Subject: Revision of manuscript ID ijerph-799869
We would like to thank to the reviewer for his comments. Please find below a point-by-point response to each comment raised by the reviewer of our manuscript. We discussed their comments and agree with most of the observations. We made the necessary revisions and included new literature to support our ideas. We strongly believe that we have improved the quality of the manuscript.
Thank you,
Alberto Jiménez-Maldonado, Ph.D.
Corresponding author
Facultad de Deportes
Universidad Autónoma de Baja California
We show the original comment/question in boldface followed by our response and the proposed changes to the manuscript in blue.
In the original manuscript we show the changes tracks in blue for any addition and red (strikethrough) for any deletion.
REVIEWER 3
Comments and Suggestions for Authors
The authors have responded to my original comments. I have a few follow up comments related to my original one.
Answer: We thank the reviewer for this comment. We have worked diligently to answer all the comments and questions from the reviewers. As a result of this work, we believe that the quality of the manuscript has improved considerably.
Comments
Line 222 – I guess the authors were lucky in that the randomization method resulted in no differences between groups for baseline characteristics.
Answer: We thank the reviewer for this comment. All participants were physically-active (Self-reported IPAQ short form: 6819.3 ± 4505.2 METs; minimum value: 1626, maximum value: 15813). In addition, the participants were majoring in Physical Education; thus, their lifestyles were fairly homogenous.
Changes: We included this information in Methods section (Ln 81-82 of the revised manuscript version).
Original comment 4 – the authors have not answered “why” they did not choose this design. It is a more powerful option and not addressed in the response.
Answer: We thank the reviewer for this comment. The crossover design is more commonly applied in clinical trials (Harris et al, 2017). However, this protocol (crossover) has some drawbacks, the main disadvantage is that the effects of the treatment may carry over and alter the response of subsequent treatments (Sibbald, 1998). If we had followed a crossover design, we should had needed a reasonable washout time and re-start the intervention all over again. This might have increased the risk for dropouts, which might have directly affected the statistical power of our study. With the research design chosen, we were able not only to study between-subject but also within-subject variability by appropriate statistical analyses.
References
Harris, J. E., & Raynor, H. A. (2017). Crossover designs in nutrition and Dietetics research. Journal of the Academy of Nutrition and Dietetics, 117(7), 1023-1030.
Sibbald, B., & Roberts, C. (1998). Understanding controlled trials crossover trials. Bmj, 316(7146), 1719-1720.
Original comment 8 – Yes, the authors are mostly correct. However, RIT did not change VO2max but it did change other parameters. Therefore, activity outside the scope of the study may have influenced results other than VO2max. The authors need to discuss this as a limitation.
Answer: We thank the reviewer for this comment. Even though we did not monitor changes in physical activity levels throughout the study, the participants had similar daily activity levels as described in subheading 2.1. In addition, both, participants on the experimental and the control group were instructed to maintain their physical activity levels unchanged. Our data indicate that the activity outside the treatment was the same in all participants, suggesting that our findings are explained by the experimental procedures (i.e., intervention). We have included this discussion in Ln 288-290 as a relevant recommendation for the readers of the journal when designing similar studies.
Original comment 10 – the authors have not considered the comment with respect to “this” study. Changes in HRrest massively influenced the outcome. Per my original comment, the authors need to discuss this matter.
Answer: We thank the reviewer for this comment. We would like to indicate that using the intra-group analysis we reduced the bias of DP change between groups analysis and demonstrate that the change in both, DP and HRrest, was affected in both CON and RIT.
Changes: We added new information in the Discussion section to address the change in DP (Ln 259-263).
All the changed text does not align with the response to reviewers. For example, comment 9, author’s state that changes in SIT SBP were discussed in 186-188 and 244-245, except there is no line 245.
Answer: We thank the reviewer for this comment. We apologize for the typo; we rephrased lines 186-188 (revised manuscript version) to clarify this issue, which was also mentioned in lines 246-250 of the current version of the manuscript.
